# Effect of Flow-Induced Shear Stress in Nanomaterial Uptake by Cells: Focus on Targeted Anti-Cancer Therapy

**DOI:** 10.3390/cancers12071916

**Published:** 2020-07-16

**Authors:** Samar Shurbaji, Gulsen G. Anlar, Essraa A. Hussein, Ahmed Elzatahry, Huseyin C. Yalcin

**Affiliations:** 1Materials Science and Technology Department, College of Arts and Sciences, Qatar University, Doha 2713, Qatar; samar.h.shurbaji1993@gmail.com (S.S.); essraa_88@yahoo.com (E.A.H.); 2College of Medicine, Department of Medical Sciences, Qatar University, Doha 2713, Qatar; guliz-y@hotmail.com; 3Biomedical Research Center, Qatar University, Doha 2713, Qatar; 4Department of Biomedical Sciences, College of Health Science-QU Health, Qatar University, Doha 2713, Qatar

**Keywords:** nanomedicine, nanoparticle, targeted therapy, anti-cancer, shear stress, flow, in vitro

## Abstract

Recently, nanomedicines have gained a great deal of attention in diverse biomedical applications, including anti-cancer therapy. Being different from normal tissue, the biophysical microenvironment of tumor cells and cancer cell mechanics should be considered for the development of nanostructures as anti-cancer agents. Throughout the last decades, many efforts devoted to investigating the distinct cancer environment and understanding the interactions between tumor cells and have been applied bio-nanomaterials. This review highlights the microenvironment of cancer cells and how it is different from that of healthy tissue. We gave special emphasis to the physiological shear stresses existing in the cancerous surroundings, since these stresses have a profound effect on cancer cell/nanoparticle interaction. Finally, this study reviews relevant examples of investigations aimed at clarifying the cellular nanoparticle uptake behavior under both static and dynamic conditions.

## 1. Introduction

In 1959, Richard Feynman delivered his pioneering lecture about nanotechnology in which he gave a foundation about materials miniaturization [1]. Since then, nano-scaled materials have been investigated and studied extensively for use in various fields, including the medical field [2]. When the power of nanotechnology is harnessed for biomedical applications, it is designated as nano-biotechnology or bio-nanotechnology to indicate the combination of nanotechnology with the biological system [3]. Nanomaterials are considered promising and favorable materials due to their unique properties as well as their extremely small size and high surface area to volume ratio, which means better surface interaction and effective cellular uptake. Nanobiotechnology has been applied in diverse medical applications, such as drug delivery platforms, contrast agents for magnetic resonance imaging, tissue engineering, and anti-cancer therapy. 

Today, cancer is rated as the second leading cause of mortality worldwide [4]. In cancer cases, the signals that control normal cell division and normal cell death are disregarded due to genetic or environmental conditions. Consequently, uncontrolled cell division gives rise to rapid cell growth and the formation lumps, which is known as localized tumors. These tumor cells are characterized by fast proliferation, metastasis, and the ability to induce the formation of new blood vessels, which is also known as “angiogenesis” [5]. Current cancer therapies are known for their lack of selectivity for tumor cells, as well as severe side effects such as damage to healthy organs, hair loss, and uncontrolled gastric problems. The integration of nano-scaled structures for anti-cancer therapy can be in the form of carriers for chemotherapeutic agents, cancer diagnostic agents, or targeting moieties. Nanomedicine holds the potential to minimize the undesired and severe adverse side effects of anti-cancer therapy, as well as to increase the efficacy and selectivity against tumor cells. In that regard, significant efforts have been devoted to developing nanoplatforms for specific cancer therapy or nanomedicine [6,7,8,9]. To design an effective nanomedicine, specific characteristics of cancer cells such as cancer cell mechanics or microenvironment of the tumor, which will influence the binding or internalization of the nanoparticles to cancer cells, should be taken into consideration. 

Cancer cells are exposed to different forces and mechanical stresses than normal cells in the body, such as compressive forces due to tumor growth plus the interstitial pressure and shear stresses due to blood and interstitial fluid flow [10]. The biophysical microenvironment of tumor cells is different from normal cells. To illustrate this, blood flow in cancer microenvironment is irregular compared to normal circulation and subsequently, causes the tumor to be less oxygenated as the tumor grows [11]. Furthermore, the tumor site (extracellular fluid) is more acidic than normal tissues [12]. All these differences have substantial influences on the interactions of tumor cell with applied nanostructures. For example, shear forces in the extracellular environment can activate some cellular processes and affect the cellular uptake mechanism, which is important for targeted cancer therapy via nanoparticles [13].

Generally, fluid shear stress (FSS) in the biological systems can be categorized as resulting from blood flow, interstitial fluid flow or lymphatic fluid flow. Cancer cells mainly encounter interstitial fluid flow in localized tumor and also blood flow in case of metastasis [14]. Tumor cells can be exposed to additional fluid flows in the body, such as fluid flow in peritoneal cavity during ovarian cancer, which increases FSS [15]. Consequently, FSS is accepted as an important factor regulating the behavior of cancer cells and, more particularly, FSS acting on tumor cells will be discussed later in this article.

The major objectives of this review are to: (a) demonstrate the main types of physiological shear stresses that are affecting the tumor cells; (b) shed light on the interactions between cancer cells and applied nanomaterials in both static and dynamic conditions; (c) summarize findings on the influence of uptake of nanomaterials by cancer cells.

## 2. Physiological Shear Stresses Affecting the Tumor Cells

### 2.1. Shear Stress Due to Blood Flow 

Circulating tumor cells (CTC) or metastatic cells are cancer cells that shed from the localized primary tumor and migrate to other body sites through the blood stream [16]. These cells experience shear stress due to blood flow [17,18]. Studies showed that, CTC can be influenced by FSS in two ways: either the cell cycle will be arrested due to mechanical force [19] or certain cellular process will be activated resulting in migration of CTC and invasion of other organs [20]. It has been reported that high levels of FSS (~60 dyn/cm^2^) can induce apoptosis and eliminate 90% of cancer cells from the blood stream [21]. This elimination and cell death have been related to destruction of the cell’s cytoskeleton due to high shear, thus preventing cell adhesion. Furthermore, at high shear rate, cells produce more reactive oxygen species, resulting in cell death due to oxidative stress [22]. On the other hand, low FSS (2 dyn/cm^2^) can activate certain mechanosensitive cytokines such as IGF-2, VEGF, ROCK, and Cav-1. This activation prompts their downstream molecular pathways which induce metastasis [23]. 

### 2.2. Shear Stress Due to Interstitial Fluid Flow 

Molecular diffusion and convection are the basic mechanism of biological mass transport. In molecular diffusion, random molecular movements lead to net transport of solutes or particles down the gradient in concentration. During convection, a solute or particle is carried by moving fluid. In a region where a fluid (for example, blood or interstitial fluid) is flowing, diffusive and convective transport can occur simultaneously [24]. 

In normal tissue, the way that cells get their nutrition is by diffusion of the blood plasma to the stromal space between the cells, which is also known as the interstitial space. The cells excrete their wastes by diffusion of waste products to the nearby lymphatics that drain them to the venous blood stream [25] (Figure 1). In normal situations, the flow of interstitial fluid is only maintained by the diffusion of nutrients from the blood stream to the interstitial space, and waste products from the cells to the interstitial space, and then to the lymphatic vessels. This mechanism prevents excessive fluid accumulation in interstitial site. However, the situation is different in the cancer microenvironment. As cancer cells keep growing, it becomes difficult for them to support a good waste drainage. Furthermore, tumor endothelial cells proliferate fast due to production of vascular endothelial growth factor (VEGF) by tumor cells. However, they form less tight junctions than endothelial cells in normal tissue, causing leaky endothelial cell junctions and hyperpermeability [26]. Therefore, although the fluid will be absorbed from the blood vessels, it will not be drained back to the venous system. This fluid accumulation will cause pressure difference between cancer microenvironment and healthy tissue, resulting in fluid flow from tumor to its surroundings [25]. The flow of interstitial fluid was shown to induce shear stresses on the cancer cells within the localized tumor [27] with a shear stress level of 0.1 dyn/cm^2^ [13]. Interstitial flow has much slower velocity than blood flow. The interstitial flow velocity ranges from 0.1–4 um/s compared to blood flow at 0.6–0.9 m/s in pulmonary artery [28]. Table 1 represents levels of FSS in the body and Figure 2 shows different types of FSS that cancer cells are exposed to.

### 2.3. Important Aspects for the Development of Nanomedicine for Targeted Cancer Therapy

One important aspect to consider is the mode of transport of drugs to cancer tumor, which is combination of convection and diffusion. Once infused, anti-cancer agent is transported in the systemic circulation via convection. Upon reaching to microcirculation, exchange occurs between blood and tissue. Here, drug passes through vessel walls toward cancer cells by combination of convection and diffusion in interstitial fluid. For low molecular mass drugs and small nanoparticles, diffusion is the dominant transport mechanism [31].

For efficient targeted anti-cancer therapy using nanoparticles, the tumor microenvironment should be considered during the design process. Ideally, nanomaterials, i.e., nanoparticles, or photothermal nano-agents should be tested on pre-clinical animal models of cancer therapy. However, using animal models is limited by ethical guidelines, also it is time and labor intensive [32]. To avoid the uncritical testing on animals, in-vitro and in-silico testing are used as preliminary evaluation due to their low cost, simplicity and better control on experimental conditions. In silico simulations are developed to analyze nanoparticle/cancer cell interactions by solving governing physical equations. These computational models provide quantitative analyses to describe biological mechanisms under certain conditions. However, in most situations, in-vitro experiments should be designed to verify in-silico test results [33]. For example, using a combination of in-vitro flow chamber set up and in-silico simulations, Boso et al. showed that artificial neural networks can determine the optimal nanoparticle size for maximal adherence to a targeted tissue. This optimal size depends on the wall shear rate in the target location [34]. The results suggested that the number of in-vitro experiments can be successfully reduced by using artificial neural networks, without compromising the accuracy of the study.

One of the major limitations for the in vitro approach is the discrepancies compared to in-vivo systems. The reason for these discrepancies is related to the fact that cells in the body are influenced by many factors in their native environment. For example, FSS is one important factor affecting cell behavior. Therefore, static cell cultures are limited to mimicking the native cancer environment. To resemble the real conditions in organized system, FSS can be induced to static cell culture by using microfluidic devices [35]. FSS is the force experienced by cells as a result of flow of viscous fluids [17]. FSS can be applied on cells using parallel plate flow chambers (PPFCs), cone plate chambers or microfluidic chambers. Different chambers are used based on the site where FSS is intended to be mimicked. For example, cone-plate chambers are used to mimic FSS in abdominal aorta and brachial artery due to resemblance of their geometry [36], whereas parallel plate or microfluidic chambers are used to mimick FSS in smaller vessels.

PPFC were commonly used to mimick FSS in cancer microenvironment since cancer cells in the body are constantly exposed to FSS by interstitial flow or blood flow. It was previously suggested that FSS is an important factor for nanoparticle internalization by cancer cells. Therefore, association of FSS and cellular uptake of some nanomaterials has been studied [37,38]. We will explain this in detail in the following section. One of the first PPFCs was developed in 1995 by Ruel et al. [39]. A typical PPFC would have an inlet port and an outlet port for flow perfusion, silicon gaskets to form the flow chamber, and a coverslip where cells are grown on (Figure 3).

These flow chambers are mostly connected to syringe or peristaltic pumps that can pump a certain fluid (mostly cell media) at specific flow rates for extended flow perfusion. Shear stress can be calculated using Hagen–Poiseuille equation assuming Newtonian fluids under steady and laminar flows.
$$\tau =\frac{6.µ.Q}{w.h^{2}}$$
where Q represents the fluid flow rate, τ is the shear stress acting on the cells, w and h are width and height of the flow chamber, and μ is the viscosity of the fluid, which is the cell medium [40]. Figure 3 illustrates a typical chamber setup representing the flow of the fluid in a closed circuit.

PPFCs offer a model that is not as simple as static cell culture, but not as complex as animal models; thus, cellular interactions and nanomaterials uptake can be studied in a practical and reliant manner as represented in Figure 4 [41].

## 3. Interactions between Nanoparticles and Cancer Cells

Nanomaterials interact with cells differently in static and dynamic cultures. These differences include production of reactive oxygen species (ROS) [43] as well as the viability and uptake of the nanomaterials by cells [44]. Dynamic culture is more relevant to physiological conditions present in an animal or human body, as the biological systems are more complex and dynamic. Usually, it is easier to study the influence of nanomaterials using static cultures, but the results from such studies might be misleading and/or contradictory when compared to animal models or dynamic cultures. For example, nanomaterials tend to sediment and settle down in static cultures, inducing stresses on cells. Furthermore, nanomaterials form aggregates in static cultures, which might alter their uptake by the cells, and therefore, altering the viability of the results explained in Figure 5 [43].

When nanomaterials form aggregates, the aggregate size should be much smaller than the cell size for uptake. There are different mechanisms by which cells uptake nanomaterials. These include diffusion or passive penetration through the plasma membrane, and endocytosis that involves pinocytosis and phagocytosis. Pinocytosis involves the internalization of molecules or fluid by the formation of small vesicles, whereas phagocytosis involves the engulfment of large materials by the formation of intracellular phagosomes [45]. It was reported that the uptake of nanomaterials is size-dependent, and in some cases, it is easier for the cells to uptake larger nanomaterials by endocytosis, than smaller nanomaterials by diffusion [46]. Moreover, the formation of aggregates and sedimentation of nanomaterials will alter the effective concentration of nanomaterials delivered to the cells [47]. Therefore, nanoparticle aggregation should be prevented in most cases for nanoparticle studies. To uniformly distribute the nanomaterials over cells in culture without aggregate formation or sedimentation, it is suggested to use dynamic culture, and grow the cells under flow conditions using flow chambers [35].

It was reported that the uptake of nanomaterials is different under flow conditions compared to static cultures and that these changes are due to material’s surface charge, surface ligands, stiffness, size and shape [48]. Cells can uptake nanomaterials in two steps: the first step is binding of the nanomaterial to cell surface and the second step is internalization of the nanoparticles. In the first step, electrostatic interactions, which are due to the physio-chemical properties of the nanomaterial, play an important role. As the cell membrane is negatively charged, it is more favorable for positively charged materials to interact with its surface than neutral or negatively charged particles. The second step is the internalization of the nanoparticle from the cell membrane. After nanomaterials interact and bind to the cell surface by electrostatic interactions, they can then be internalized by different uptake mechanisms [49]. Although surface charge is considered as an important contributor to higher uptake, other parameters influence the cellular uptake as well, such as elasticity [50] and the shape of the material especially under flow conditions [51]. Under flow conditions, the alignment of non-spherical nanomaterials can be different from that in static culture, thus altering the uptake. It has been reported that fibrous or 2D materials have a flow-aligning effect, which impacts their cellular adhesion and uptake [41], as demonstrated in Figure 6.

## 4. Shear Stresses and Cellular Uptake of Nanomaterials for Cancer and Normal Cells

Owing to the effect of the dynamic environment in different biological processes, tissue engineering, and drug-delivery [52,53], many studies have investigated the role of FSS in the interactions between cells and nanoparticles [54,55,56]. One of these significant interactions is the cellular uptake of nanoparticles. To illustrate this, the uptake of the applied nanomaterials by cells is considered an important aspect, especially in drug-delivery and other therapeutic purposes, which require sufficient uptake by the targeted tissue. One important factor playing a role in cell-nanomaterial interaction under flow is the surface e charge of the nanoparticles. For instance, the interaction of endothelial cells with two negatively charged nanoparticles has been scrutinized by Samuel et al. by the application of varying levels of FSS on cells [57]. The authors revealed that, the cellular uptake increased under low shear stresses (0.05 Pa) compared to high shear stresses (0.5 Pa). In static conditions (0 Pa), cellular uptake was lower compared to low shear stress (0.05). The higher uptake of these particles under stress was mainly attributed to the formation of cytoskeletal stress fibers and membrane ruffles, which enhance endocytosis. Such changes in the cytoskeleton were not observed in the non-shear exposed cells. Additionally, Rigau and Städler correlated between the uptake of nano-sized drug delivery systems and the subsequent therapeutic effect using skeletal mouse myoblast cell model (C2C12) in the absence or presence of FSS [58]. They concluded that, the liposomes with positively charged lipids result in higher cellular interaction in the presence of shear, in contrast to those contained negatively charged lipids or zwitterionic ones. Furthermore, the authors investigated the therapeutic effect, in terms of cell viability, after treatment with the positively charged liposomes carrying a small cytotoxic molecule in static and dynamic conditions. Their findings stated that, there was a higher therapeutic response (i.e., higher cell mortality) in the case of dynamic conditions, which demonstrates the relationship between the higher cellular association of positive carriers and more effective therapy in the presence of shear. In another relevant study by Rinkenauer et al., authors investigated the effect of FSS on the uptake of co-polymers (negatively charged PMMA -co-PMAA with different ratios of MAA (3%, 5%, 8%, and 13%) and positively charged PMMA-co-PDMAEMA with 20% PDMAEMA) using different cell lines (HUVEC, HEK293, L929, and primary muscle cells). They found that, increasing the negative charge (MMA) increases the uptake by different cells under static conditions. However, the uptake is not as efficient as that resulting from the use of positive particles (20 % PDMAEMA). A similar trend was observed in different cell lines, but not in co-culture which reduced the cellular uptake due to cellular interactions. When the uptake was assessed under flow conditions (0.7, 3, 6, and 10 Dyn/cm^2^), it was observed that, increasing shear stress is positively correlated with cellular uptake. Nevertheless, compared to static culture, 13% PMMA showed more efficient uptake compared to positively charged 20% PDMAEMA. This is probably related to the differences in surface receptor patterns observed under flow conditions, which can alter the cellular uptake [38]. In our group, we are developing two-dimensional Mxene sheets as photothermal agents. Our initial findings show successful uptake on MXene sheets by MDA 231 breast cancer cells. When we compared static and dynamic cultures, we did not see any significant difference.

Another important aspect in cell-nanoparticle interaction is the surface modification. Toe et al. studied the cell response to modified liposomes with and without FSS using two cell lines. The former cell line was the immortalized skeletal mouse myoblast (C2C12), a tumor cell model, which is important to estimate the activity of the applied liposomes as drug carriers in drug delivery systems. The later cell model was hepatic cells (HepG2), which was chosen as a model for hepatic clearance due to their importance in eliminating drug-loaded nanocarriers from the body. To illustrate, the authors fabricated PEGylated poly (dopamine) coated liposomes and quantified their cellular uptake by myoblasts and hepatocytes using flow cytometry in both static and dynamic conditions. The results manifested that the hepatocytes response in the dynamic conditions was significantly higher after only 30 minutes, while the myoblasts demonstrated a significant increase after a relatively longer time (4 h). The authors explained these findings as the nature of the two cell lines were different. The hepatic cells were concerned with clearance, so their responses were instantaneous in the presence of physiological shear. On the other hand, the cancer cell model needed a longer time to show a response in low shear stress (0.146 dyn/cm^2^) [59]. Additionally, the uptake of lipidic NPs by MCF-7 breast cancer cells and Hela human cervical cancer cells was reported by Palchetti et al., under flow conditions. Authors produced two types of lipidic NPs, one with surface modification (PEGylated) while the other without modification. They incubated the cells with particles at two different incubation durations (5 and 90 minutes). MCF-7 cells showed a significantly lower uptake of unmodified NPs in dynamic culture in comparison to static condition at both incubation durations, whereas Hela cells showed a higher NPs cellular uptake after 90 minutes incubation in dynamic culture [53]. On the other hand, an insignificant difference in uptake of modified NPs by MCF-7 was observed under flow and static conditions. However, NPs uptake by Hela cells in dynamic conditions was still higher than static culture. They clarified that shear stress can affect the protein corona (protein corona is formed when NPs absorb biomolecules as they interact with cells and the biological system) by changing its surface chemistry and properties, which in turn affect their uptake by cells [60].

Particle elasticity is suggested to affect cellular uptake [61,62]. In a very important investigation, Guo et al. revealed experimental evidence that indicates how elasticity alters in-vitro cellular uptake and in-vivo tumor uptake [50] They studied uptake of nanolipogels (NLGs) which consist of lipid bilayer capsule and hydrogel core with tunable elasticity. The elasticity of NLGs could be modulated independent from other physical properties such as size, shape and surface charge. Both normal cells and cancer cells showed higher uptake of soft NLGs (NLP-45KPa) compared to rigid NLGs (NLG-19MPa). Authors explained the higher uptake of soft particles by usage of different cell internalization pathways. While NLP-45KPa entered the cells through fusion and endocytosis, NLG-19MPa was internalized by only endocytosis (Figure 7). Fusion requires low energy compared to endocytosis. Therefore, cells take more time and energy to uptake the same amount of NLG-19MPa than NLP-45KPa. The in-vivo test results showed that particles with higher elasticity were more likely to accumulate into tumors. This is strong evidence that particle stiffness controls the tumor uptake of systematically applied nanoparticles.

Surface ligand is another aspect that affects cellular intake. Several studies were conducted on selective tumor targeting in order to eradicate tumor cells without harming normal body cells. The experiments were based on decorating nanoparticles with molecular recognition ligands that bind to selective proteins expressed on the surface of cancer cells. Engelberg et al. studied the internalization of quantum dots (QDs) decorated with S15-APT ligand into human non-small cell lung cancer A549 cells [63,64]. S15-APTs is a selective targeting moiety for uptake by A549 cells. These APT-decorated QDs bound themselves selectively to the target A549 cells and were internalized by them. However, they were neither bound to, nor were internalized by normal human bronchial epithelial BEAS2B, cervical carcinoma (HeLa), and colon adenocarcinoma CaCo-2 cells, thereby demonstrating high specificity. The shape and size of the particle is known to affect the uptake as well. Particle shape- and size-dependent uptake under physiological shear stress was reported by Jurney et al. They produced negatively charged rod-shaped PEG NPs with different aspect ratios and assessed their uptake by human umbilical vein endothelial cells (HUVEC) under flow conditions at different incubation durations (1, 12, and 24 hours). In all cases, the uptake of larger particles was found to be higher than smaller ones under flow in comparison to static culture. In contrast, smaller particles are internalized more in static conditions than in flow conditions. The trend of larger NPs being internalized more under flow conditions is contradictory with what was reported in literature with similar-sized spherical NPs. This indicates that particles with higher aspect ratios interact more with cells under flow conditions [65].

Moreover, Klingberg and Oddershede studied the effect of FSS on the uptake of spherical 80 nm gold nanoparticles (Au NPs) by HUVEC [66]. They categorized the cells into two groups, one group was cultured in static conditions for 24 hours (non-adapted group), while the other group was cultured for 24 hours under 10 dyn/cm^2^ shear stress (shear adapted group). Then, each group was either kept in static culture for three hours in the presence of 5 µg/mL Au NPs or kept in dynamic culture for three hours in presence of 5 µg/mL Au NPs. The highest uptake was achieved by non-adapted group with three hours additional static culture and lowest uptake was realized by the shear adapted group with three hours additional dynamic culture. [66]. One more study was conducted by Fede et al. to reveal the effect of FSS and size of spherical citrate stabilized gold nanoparticles on HUVEC [32]. They tested two batches of gold NPs (Batch 24 nm and Batch 13 nm). It was observed that, the viability is significantly more when testing gold NPs under flow conditions in comparison to static culture, regardless of NPs size or concentration. They measured the NPs concentration in two methods, one based on the surface area per unit volume, while the other based on the number of NPs per unit volume. They found that the cells were less viable when the surface area was increased per unit volume irrespective of the NPs size [32]. Yazdimamaghani et al. studied the effect of silica NPs density and flow conditions on cell cytotoxicity, uptake and sedimentation. They produced four types of silica NPs with different densities and surface charges and tested the cytotoxicity and uptake on RAW 264.7 macrophage cells after 24 hours of incubation with the cells, in static or under flow conditions. They found that the cell viability is enhanced under flow conditions, compared to static culture. Moreover, none of the four particles showed a toxic effect on macrophage cells up to 250 µg/mL in dynamic conditions. Also, particle sedimentation was reduced in dynamic conditions, and the distribution of particles was more homogeneous. Authors also found that, cellular uptake of silica NPs was more in static conditions compared to dynamic conditions. Furthermore, low density particles, showed lower uptake under flow conditions compared to high density particles [67].

Finally, application of different FSS levels is considered as an important parameter for detailed investigation of the effect of dynamic conditions on cellular responses, where the cellular uptake could be studied as a consequence of all applied FSS levels. Hence, more relevant correlation between the cell response and FSS levels can be stated. For instance, Kona and co-workers developed a novel drug delivery system that imitates the natural platelet adhesion to the injured vascular walls under different shear flow rates [68]. Their results implied that when the shear stress level was increased to 20dyn/cm^2^, the cellular uptake fell dramatically by three folds when compared to the control static group. The authors explained their findings through computational model revealing that the high shear rates induce huge dislodging forces that are able to detach the adhered particles [69,70]. Table 2 summarizes important works on the effect of FSS nanoparticle internalization.

## 5. Conclusions

Cancer is a wide spreading disease with no definitive treatment. Researchers have been working on cancer therapy for decades with some improvements, yet many limitations remain. Lately, nanomaterials are being used for various biomedical applications including the targeted anti-cancer therapy due to their superior properties. Usually, when nanomaterials are tested for biomedical applications, cell culture techniques are used for preliminary testing. Cell culture is the most convenient method to test the toxicity and efficacy of nanomaterials, but it is limited due to particle aggregation, sedimentation and it does not mimic the native conditions in animal model and human body. FSS is one important parameter that affect nanomaterial-cell interaction, mainly cell viability and particle uptake. FSS can be due to blood flow, with variable flow rates based on the diameter size of the blood vessel where it affects endothelial cells lining the blood vessels or the circulating tumor cells. FSS can be due to interstitial fluid flow as well with very low flow rate, which occurs mainly around cancer cells in solid tumors. Here we summarized findings on the relation between shear stress and nanomaterials uptake mainly for cancer as well as for normal cells using in-vitro systems. There are variety of factors affecting nanomaterials uptake particularly under dynamic conditions. Some of these factors are related to the nanomaterials, while other factors are cell related. Nanomaterial size, shape surface charge, surface ligands, and particle elasticity are the main factors in cellular uptake under fluid flow. However, these factors are affecting nanomaterial-cell interaction differently depending on the cell type (i.e., origin of tissue and cancer vs healthy). There is no general rule on how nanomaterials will interact with cells. However, in most of the cases, negatively charged particles show less uptake by cells due to inefficient electrostatic interactions between nanomaterials and cells. Furthermore, soft particles show more uptake than rigid particles which can be attributed to the ability of the cell to uptake soft particles by different pathways compared to rigid particles. Additionally, the uptake of 2D materials will be different under flow conditions due to the effect of flow aligninment. Coating the cell surface with ligands is an efficient way to guarantee the uptake of particles, at the same time reducing the side effects by preventing internalization by non-cancerous cells. Other factors that might affect cellular uptake, are cell related. For example, the cytoskeletal structure and the formation of membrane ruffles after flow, as well as, cell rigidity under dynamic culture. However, these details are not the main focus of this paper. Further investigations will shed light on optimal nanoparticle parameters that can be used as smart nanoparticles for anti-cancer therapies.

## Figures and Tables

**Figure 1 cancers-12-01916-f001:**
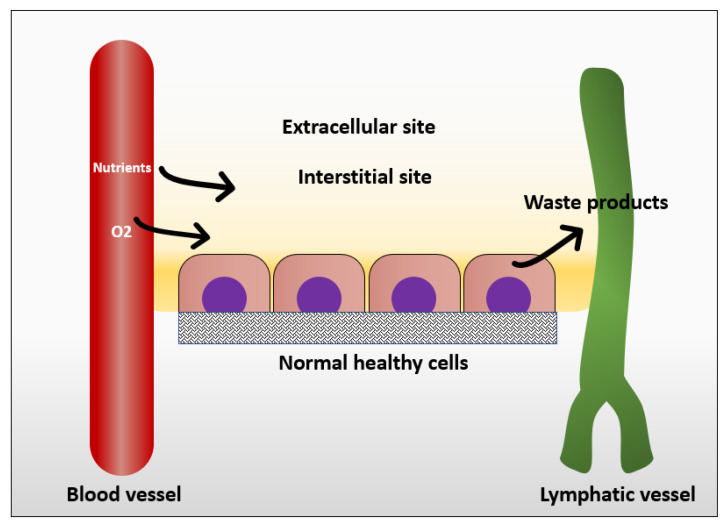
Mechanism in which normal cells get their nutrients and excrete their wastes.

**Figure 2 cancers-12-01916-f002:**
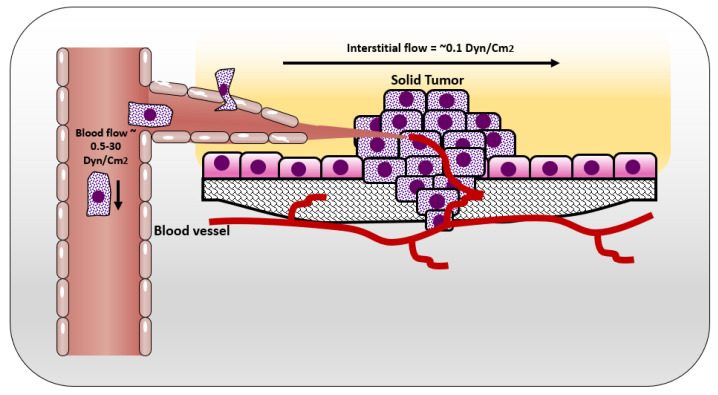
Shear stresses experienced by cells in solid tumor and circulating tumor cells.

**Figure 3 cancers-12-01916-f003:**
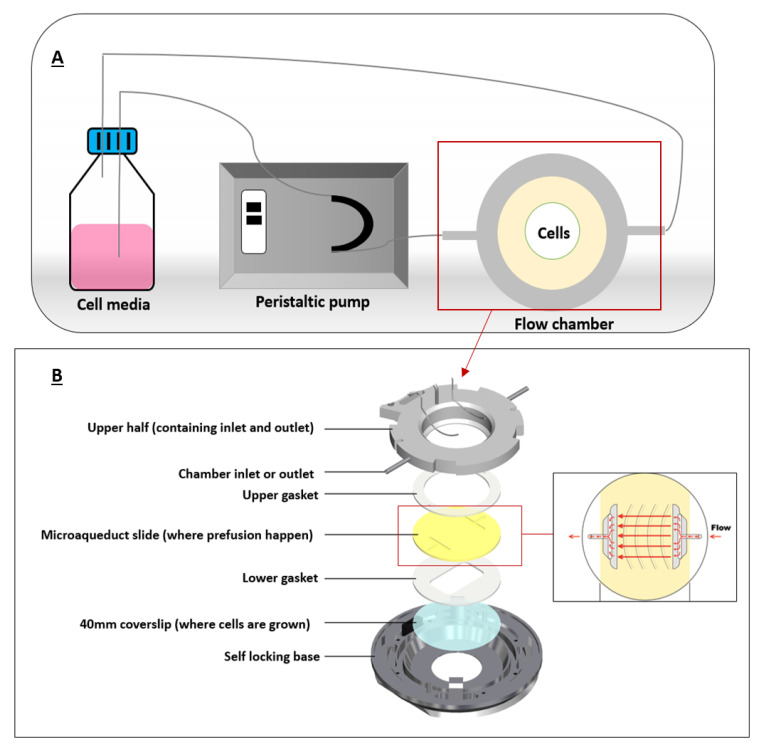
A typical flow chamber setup. (**A**) depicts a closed-circuit chamber, in which the chamber is connected to a peristaltic pump and a reservoir (cell media). (**B**) illustrates the flow chamber assembly where coverslip containing the cells is allowed for fluid flow. Adapted from bioptechs [42].

**Figure 4 cancers-12-01916-f004:**
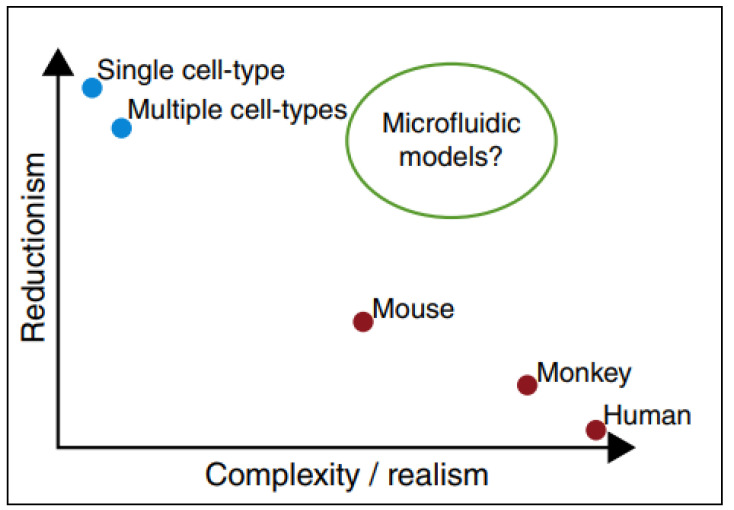
Microfluidic devices as models in which they provide conditions similar to in-vivo animal models and in-vivo models still retaining the simplicity of in-vitro testing. Adapted from Björnmalm et al. [41].

**Figure 5 cancers-12-01916-f005:**
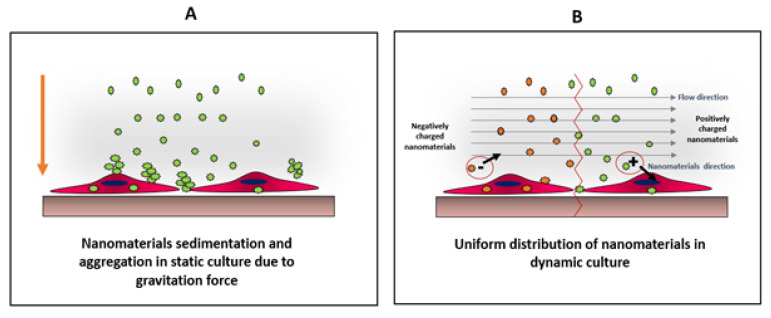
Comparison between distributions of nanomaterials under static (**A**) and dynamic (**B**) conditions. In static culture, nanoparticles tend to sediment and aggregate due to their high surface energy. This condition create physiochemical stress on cells, which might alter cells viability as well as particles uptake On the other hand, in dynamic culture, the particles will be uniformly distributed allowing better cellular interaction, which can be charge-dependent as the direction of the negatively-charged particles will be away from the cell surface, unlike positively-charged particles, where the particle direction will be towards the cell surface. Adapted from Mahto et al. [43].

**Figure 6 cancers-12-01916-f006:**
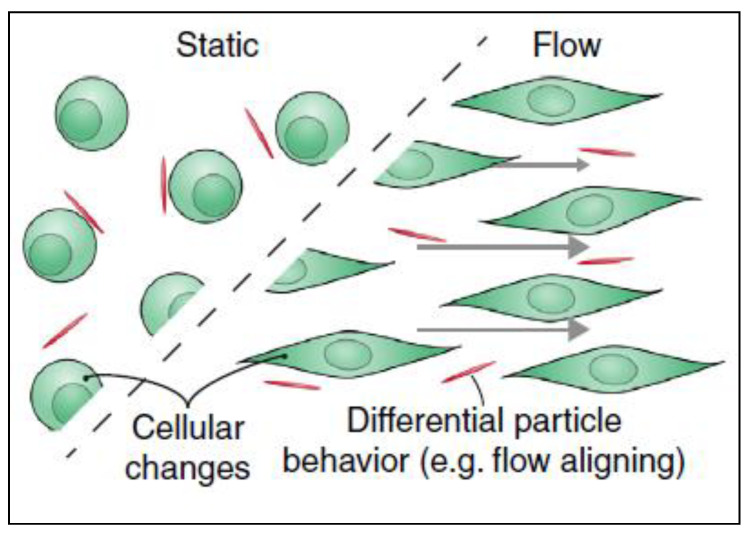
Filamentations or 2D nanomaterials align differently when there is fluid flow in cancer microenvironment. This flow-aligning effect can change the way that the cells interact with nanoparticles thus their cellular uptake would be influenced. Adapted from Björnmalm et al. [41].

**Figure 7 cancers-12-01916-f007:**
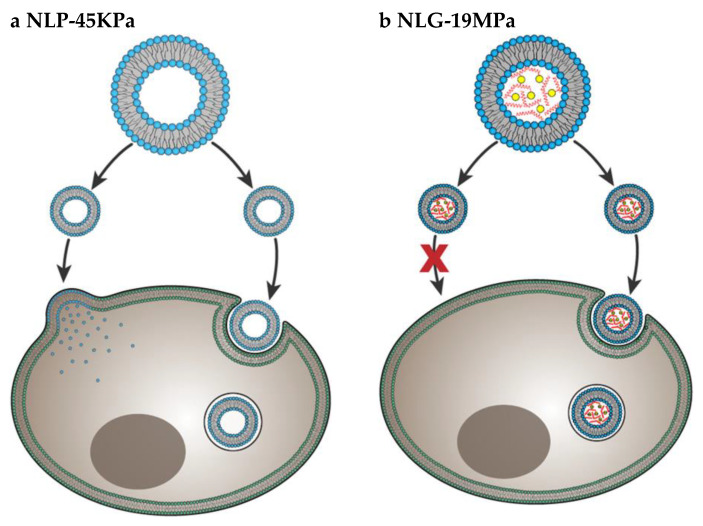
Cell internalization pathways of particles with different elasticity. Soft NLP-45KPa (**a**) enters the cell via two pathways: fusion and endocytosis. Hard NLG-19MPa (**b**) enters cell via only endocytosis. Adapted from Guo et al. [50].

**Table 1 cancers-12-01916-t001:** Different shear rate values in physiological and pathological conditions.

Fluid Flow	Shear Stress (Dyn/cm^2^)	Reference
Interstitial flow	0.1	[13]
Normal vein	1–6	[29]
Normal artery	10–70	[29]
Lymphatic fluid flow	0.64	[14]
Liver	0.1–0.5	[30]
Peritoneal fluid flow	<5	[15]

**Table 2 cancers-12-01916-t002:** Summary of findings on nanoparticle—cell interactions under shear stress.

Type & Properties of Nanomaterial Applied	Type of Cells	Flow Conditions	Shear Rate	Findings	Ref.
Different polymer-based NPs	HUVEC	One-hour incubation under optimum conditions (37 °C, 95% air & 5% CO2)	0.7, 3, 6, and 10 Dyn/cm^2^	Increasing the negative charge increases the uptake under static conditions. Positively charged particles showed more efficient uptake in static culture compared to negatively charged particles. Increasing shear stress is positively correlated with cellular uptake.	[38]
PEGylated lipidic NPs. Un-PEGylated lipidic NPs	MCF-7 Hela cells	Incubated under optimum conditions for 5 or 90 minutes under flow speed of 50 cm/s		MCF-7 cells showed significantly lower uptake of un-PEGylated NPs in dynamic culture at both incubation durations. PEGylated NPs showed similar uptake by MCF-7 in dynamic culture and static culture. Hela cells showed a higher NPs cellular uptake after 90 minutes incubation in dynamic culture.	[60]
Negatively charged PEG NPs with different aspect ratios	HUVEC	1,12, and 24 hours of exposure to dynamic conditions using 0.907 uL/min flow rate.	10 Dyn/cm^2^	Larger particles have higher internalization than smaller ones, under flow in comparison to static culture.	[65]
Silica NPs with different densities	RAW 264.7 macrophage cells	24-hour incubation under flow (cell media agitation) at optimum conditions.		More Uptake in static conditions compared to dynamic conditions. Low density particles have lower uptake in dynamic conditions compared to high density particles.	[67]
Negatively charged NPs	Endothelial cells	24-hour incubation under optimum conditions.	0.05, 0.1, and 0.5 Pa	cellular uptake increased with low shear stresses when compared to high shear.	[57]
Negative, Positive & Zwitterionic lipidic NPS	Skeletal mouse myoblast cell model (C2C12)		0.0146 and 0.146 Dyn/cm^2^	Higher cellular interaction in the presence of shear for positively-charged NPs compared to negatively-charged lipids or zwitterionic ones.	[58]

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
