# Peer review of "Effect of Flow-Induced Shear Stress in Nanomaterial Uptake by Cells: Focus on Targeted Anti-Cancer Therapy"

_cancers, 2020, doi:10.3390/cancers12071916_

Round 1

Reviewer 1 Report

This review manuscript gives us a summary on the relation between shear stress and nanomaterials uptake mainly for cancer as well as for normal cells using in vitro systems. Nanomaterial size, shape and surface charge are the main factors in cellular uptake under fluid flow. The manuscript is well organized and flows well. However, this review manuscript narrowly focused on the factors affecting nanomaterial uptake under different flow shear stress. Besides the cellular uptake of nanomaterials, there are some additional points to be discussed regarding the relation between the flow shear stress and cancer nanomedicine.

The stability of cancer nanomedicine is an important consideration factor when developing a nanomedicine for anti-cancer therapy. How is flow shear stress affecting the stability of cancer medicine? What is the implication of studying the relation between nanomaterial uptake and flow shear stress? Is there a direct correlation between cellular uptake and efficacy? Is there an effective way to tune flow shear stress to promote cancer therapeutic efficacy of nanomedicine? The authors are recommended to discuss more on the potential applications of this study. What are the current limitations and future prospects? There are several minor format mistakes, such as the legends for Figure 4, Figure 5 and Table 2.

Author Response

We appreciate the invaluable comments by the reviewers for our manuscript.

This review manuscript gives us a summary on the relation between shear stress and nanomaterials uptake mainly for cancer as well as for normal cells using in vitro systems. Nanomaterial size, shape and surface charge are the main factors in cellular uptake under fluid flow. The manuscript is well organized and flows well. However, this review manuscript narrowly focused on the factors affecting nanomaterial uptake under different flow shear stress. Besides the cellular uptake of nanomaterials, there are some additional points to be discussed regarding the relation between the flow shear stress and cancer nanomedicine.

The stability of cancer nanomedicine is an important consideration factor when developing a nanomedicine for anti-cancer therapy. How is flow shear stress affecting the stability of cancer medicine? What is the implication of studying the relation between nanomaterial uptake and flow shear stress? Is there a direct correlation between cellular uptake and efficacy? Is there an effective way to tune flow shear stress to promote cancer therapeutic efficacy of nanomedicine? The authors are recommended to discuss more on the potential applications of this study. What are the current limitations and future prospects? There are several minor format mistakes, such as the legends for Figure 4, Figure 5 and Table 2.

Response: As reviewer pointed out, flow shear stress is a major factor for nanoparticle uptake by cancer cells. There also other parameters of importance mainly about the characteristics of nanoparticle such as surface charge, size, shape etc. and these factors also affect uptake. According to what has been reported, for assessment of uptake, nanoparticle characteristics need to be considered along with the shear stress. However, there are not direct correlations as summarized in the manuscript. To enrich the text, we have added multiple important points for nanomedicine uptake. These were added to section 4. These include particle elasticity and surface ligands. We have fixed the format mistakes highlighted by the reviewer. The manuscript was reviewed thoroughly by a native speaker for grammatical errors. We expanded conclusion and future directions as suggested.

Reviewer 2 Report

The review paper discusses the biophysical environment of tumours and the  influence of shear stress  on cancer cell/nanoparticle interaction. First of all for a review paper it is rather incomplete. Shear stress involves mechanical aspects and these are missing or poorly described. For instance for the two sections on shear stress due to blood flow and due to interstitial flow the authors should consult the review paper by Dewhirst and Secomb (2017) for a better and more complete narrative. This is particularly important for the respective merits of diffusion versus advection.

Further when talking about testing  (lines 105  and following) the authors mention in vivo and in vitro testing but forget completely  in silico testing which often avoids or complements both previously mentioned testing methods. For this the authors should consult the recent review paper by Mascheroni nd Schrefler (2018) and the references therein. For instance Numerical methods and Artificial Intelligence can  give a great help in optimizing particle size and/or shape for best adhesion, an important aspect in drug delivery, see Boso et al. (2011). When discussing the mechanical and electrical aspects of Nano-Particles, such as size, shape, surface charges, the authors forget about stiffness which can vary between cell-like (5kPA) and rigid  (>1 Mpa). This certainly also affects the uptake, see e.g. Guo et al. (2018). Also the surface ligands (density and type) have been omitted.

As to the form: the sentences

In Figure 5 (? Figure number missing) : “In  static culture, the nanoparticles sediments due to gravitational force thus creating physiochemical stress on  the cells also sedimentation lead to aggregate formation which lead to misleading results in terms of toxicity and cellular uptake” sounds rather awkward. The same is true for the sentence on line 248: “While the latter cell  model is a liver cell line (HepG2), was chosen to investigate the efficiency of hepatic clearance of the drug  delivery nanoplatforms from the body.”

Minor remarks: a unified style for citations should be chosen, not a mix between numbers and names.

A typo: Figure 3 “adopted” instead of adapted.

In conclusion, the paper seems hastily written, is incomplete and should be seriously reworked before publication.

Author Response

We appreciate the invaluable comments by the reviewers for our manuscript

The review paper discusses the biophysical environment of tumours and the  influence of shear stress  on cancer cell/nanoparticle interaction. First of all for a review paper it is rather incomplete. Shear stress involves mechanical aspects and these are missing or poorly described. For instance for the two sections on shear stress due to blood flow and due to interstitial flow the authors should consult the review paper by Dewhirst and Secomb (2017) for a better and more complete narrative. This is particularly important for the respective merits of diffusion versus advection.

Response: As the reviewer pointed out, in addition to diffusion, convection is another transport mode influencing cancer cells. We have benefited from the paper suggested by the reviewer, Dewhirst and Secomb (2017) and also another important paper on the topic Swabb (1974) and added the following text to the manuscript: In Section 2.2, “Molecular diffusion and convection are the basic mechanism of biological mass transport.  In molecular diffusion, random molecular movements lead to net transport of solutes or particles down the gradient in concentration. During convection, a solute or particle is carried by moving fluid. In a region where a fluid (for example, blood or interstitial fluid) is flowing, diffusive and convective transport can occur simultaneously”. In section 2.3, “One important aspect to consider is the mode of transport of drugs to cancer tumor, which is combination of convection and diffusion. Once infused, anti-cancer agent is transported in the systemic circulation via convection. Upon reaching to microcirculation, exchange occurs between blood and tissue. Here, drug passes through vessel walls toward cancer cells by combination of convection and diffusion in interstitial fluid. For low molecular mass drugs and small nanoparticles, diffusion is the dominant transport mechanism”.

Further when talking about testing  (lines 105  and following) the authors mention in vivo and in vitro testing but forget completely  in silico testing which often avoids or complements both previously mentioned testing methods. For this the authors should consult the recent review paper by Mascheroni and Schrefler (2018) and the references therein. For instance Numerical methods and Artificial Intelligence can  give a great help in optimizing particle size and/or shape for best adhesion, an important aspect in drug delivery, see Boso et al. (2011). When discussing the mechanical and electrical aspects of Nano-Particles, such as size, shape, surface charges, the authors forget about stiffness which can vary between cell-like (5kPA) and rigid  (>1 Mpa). This certainly also affects the uptake, see e.g. Guo et al. (2018). Also the surface ligands (density and type) have been omitted.

Response: We thank the reviewer for these points. In-silico testing is certainly very important in nanomedicine research. We have incorporated the suggested papers, Mascheroni and Schrefler (2018) and  Boso et al. (2011) in the manuscript. Following text was added to Section 2.3:” In silico simulations are developed to analyze nanoparticle/cancer cell interactions by solving governing physical equations. These computational models provide quantitative analyses to describe biological mechanisms under certain conditions. However, in most situations, in-vitro experiments should be designed to verify in-silico test results (Mascheroni and Schrefler, 2018 ). For example, using a combination of in-vitro flow chamber set up and in-silico simulations, Boso et al showed that, artificial neural networks can determine the optimal nanoparticle size for maximal adherence to a targeted tissue. This optimal size depends on the wall shear rate in the target location (Boso et al. 2011 ). The results suggested that the number of in-vitro experiments can be successfully reduced by using artificial neural networks, without compromising the accuracy of the study”.

When discussing the mechanical and electrical aspects of Nano-Particles, such as size, shape, surface charges, the authors forget about stiffness which can vary between cell-like (5kPA) and rigid  (>1 Mpa). This certainly also affects the uptake, see e.g. Guo et al. (2018). Also the surface ligands (density and type) have been omitted.

Response: Nanoparticle elasticity is a certainly important aspect. Along with the papers suggested by the reviewer, Guo et al (2018), we have added other important papers, Liu et al.,(2012); Anselmo et al.,(2015) to the manuscript. Following text was added to section 4:” Particle elasticity is suggested to affect cellular uptake (Liu et al.,2012; Anselmo et al.,2015). In a very important investigation, Guo et al revealed an experimental evidence that indicates how elasticity alters in-vitro cellular uptake and in-vivo tumor uptake (Guo 2018 ). They studied uptake of nanolipogels (NLGs) which consist of lipid bilayer capsule and hydrogel core with tunable elasticity. The elasticity of NLGs could be modulated independent from other physical properties such as size, shape and surface charge. Both normal cells and cancer cells showed higher uptake of soft NLGs (NLP-45KPa) compared to rigid NLGs (NLG-19MPa). Authors explained the higher uptake of soft particles by usage of different cell internalization pathways. While NLP-45KPa entered the cells through fusion and endocytosis, NLG-19MPa was internalized by only endocytosis (Figure 7). Fusion requires low-energy compared to endocytosis. Therefore, cells take more time and energy to uptake the same amount of NLG-19MPa than NLP-45KPa. The in-vivo test results showed that particles with higher elasticity were more likely to accumulate into tumors. This is a strong evidence that particle stiffness controls tumor uptake of systematically applied nanoparticles”.

In Figure 5 (? Figure number missing)

Response: Fixed

 “In  static culture, the nanoparticles sediments due to gravitational force thus creating physiochemical stress on  the cells also sedimentation lead to aggregate formation which lead to misleading results in terms of toxicity and cellular uptake” sounds rather awkward. The same is true for the sentence on line 248: “While the latter cell  model is a liver cell line (HepG2), was chosen to investigate the efficiency of hepatic clearance of the drug  delivery nanoplatforms from the body.”

Response: The sentences were reworded. The sentence in Figure 5 was corrected as follows:” In static culture, the nanoparticles sediment due to gravitational force. This sedimentation creates physiochemical stress on the cells, and also results in aggregate formation that leads to misleading results in terms of toxicity and cellular uptake”. The other sentence was corrected as follows: “In static culture, nanoparticles tend to sediment and aggregate due to their high surface energy. This condition create physiochemical stress on cells, which might alter cells viability as well as particles uptake.”

Minor remarks: a unified style for citations should be chosen, not a mix between numbers and names.

Response: We unified reference style as numbers throughout the text

A typo: Figure 3 “adopted” instead of adapted.

Response: fixed

In conclusion, the paper seems hastily written, is incomplete and should be seriously reworked before publication.

Response: Conclusion was expanded. Also the manuscript went through a through grammatical review by a native speaker.

Reviewer 3 Report

Very nice paper, however the reference list needs to checked carefully, because some of the references are incomplete!

Author Response

We appreciate the invaluable comments by the reviewers for our manuscript. 

Very nice paper, however the reference list needs to checked carefully, because some of the references are incomplete!

Response: We fixed the reference list and added missing information.

Round 2

Reviewer 1 Report

The authors have addressed all the points I raised. I would like to recommend publication with no hesitation.

Reviewer 2 Report

The authors have complied wity the requests of the reviewer and the paper can now be published